# A matter of timing: Biting by malaria-infected *Anopheles* mosquitoes and the use of interventions during the night in rural south-eastern Tanzania

Isaac Haggai Namango[1,2]*, Sarah J. Moore[1,2,3,4], Carly Marshall[3,5], Adam Saddler[1,3,6], David Kaftan[3,7], Frank Chelestino Tenywa[1,2,3], Noely Makungwa[3], Alex J. Limwagu[8], Salum Mapua[8,9], Olukayode G. Odufuwa[1,2,3,10], Godfrey Ligema[3†], Hassan Ngonyani[3], Isaya Matanila[3], Jameel Bharmal[11], Jason Moore[3], Marceline Finda[8,12], Fredros Okumu[4,8,12,13], Manuel W. Hetzel[1,2], Amanda Ross[1,2]

1 Department of Epidemiology and Public Health, Swiss Tropical and Public Health Institute, Allschwil, Switzerland, 2 University of Basel, Basel, Switzerland, 3 Vector Control Product Testing Unit, Ifakara Health Institute, Bagamoyo, Tanzania, 4 School of Life Science and Biotechnology, Nelson Mandela African Institute of Science and Technology, Arusha, Tanzania, 5 British Columbia Centre for Excellence in HIV/AIDS, British Columbia, Vancouver, Canada, 6 Telethon Kids Institute, Perth, Australia, 7 New York University Grossman School of Medicine, New York, New York, United States of America, 8 Environmental Health and Ecological Sciences Department, Ifakara Health Institute, Ifakara, Tanzania, 9 University of Salford, Manchester, United Kingdom, 10 London School of Hygiene and Tropical Medicine, London, United Kingdom, 11 Innovative Vector Control Consortium, Dar es Salaam, Tanzania, 12 Faculty of Health Sciences, School of Public Health, University of the Witwatersrand, Johannesburg, South Africa, 13 School of Biodiversity, One Health and Veterinary Medicine, University of Glasgow, Glasgow, United Kingdom

† Deceased.
* isaac.namango@swisstph.ch

**Data Availability Statement:** The data and code for analysis are available from: https://github.com/

## Abstract

Knowing when and where infected mosquitoes bite is required for estimating accurate measures of malaria risk, assessing outdoor exposure, and designing intervention strategies. This study combines secondary analyses of a human behaviour survey and an entomological survey carried out in the same area to estimate human exposure to malaria-infected *Anopheles* mosquitoes throughout the night in rural villages in south-eastern Tanzania. Mosquitoes were collected hourly from 6PM to 6AM indoors and outdoors by human landing catches in 2019, and tested for *Plasmodium falciparum* sporozoite infections using ELISA. In nearby villages, a trained member in each selected household recorded the whereabouts and activities of the household members from 6PM to 6AM in 2016 and 2017. Vector control use was high: 99% of individuals were reported to use insecticide-treated nets and a recent trial of indoor residual spraying had achieved 80% coverage. The risk of being bitten by infected mosquitoes outdoors, indoors in bed, and indoors but not in bed, and use of mosquito nets was estimated for each hour of the night. Individuals were mainly outdoors before 9PM, and mainly indoors between 10PM and 5AM. The main malaria vectors caught were *Anopheles funestus* sensu stricto and *An. arabiensis*. Biting rates were higher in the night compared to the evening or early morning. Due to the high use of ITNs, an estimated 85% (95% CI 81%, 88%) of all exposure in children below school age and 76% (71%, 81%) in older household members could potentially be averted by ITNs under current use patterns.

rossaman4/timing_infected_bites_human_behaviour.

**Funding:** IHN was supported by a Swiss Government Excellence Scholarship. SJM was supported in part by the generous support of the American people through the United States Agency for International Development (USAID). The "Accelerate to Eliminate Malaria" program is a five-year cooperative agreement funded by the U.S. Agency for International Development under Agreement No. 7200AA23CA000025, beginning October 2023. It is implemented by the Innovative Vector Control Consortium (IVCC). The information provided in this document is not official U.S. Government information and does not necessarily represent the views or positions of the U.S. Agency for International Development.

**Competing interests:** The authors have declared that no competing interests exist.

Outdoor exposure accounted for an estimated 11% (8%, 15%) of infective bites in children below school age and 17% (13%, 22%) in older individuals. Maintaining high levels of ITN access, use and effectiveness remains important for reducing malaria transmission in this area. Interventions against outdoor exposure would provide additional protection.

## Background

Vector control interventions protect people by reducing or preventing human-vector contact [1]. Between 2000 and 2021 an estimated two billion cases and 12 million deaths were averted by malaria control programmes [2]. A substantial proportion of the cases averted have been attributed to the most widely used measures against malaria-transmitting *Anopheles* mosquitoes, insecticide-treated nets (ITNs) and indoor residual spraying (IRS) of insecticides [3,4]. These tools remain a critical part of the global malaria control and elimination agenda [2,5]. However, the gains made by vector control are being undermined by multiple factors, among them, insecticide resistance [6–8], sub-optimal bioefficacy and sub-optimal durability of nets [9,10], inefficient distribution of nets to households, unequal allocation of nets to household members and the poor use of nets [11,12]. In many settings, the effectiveness of indoor interventions is also attenuated by mosquito behavioural adaptations, such as biting alternative hosts, exiting houses immediately after feeding to rest outdoors and biting humans outdoors [13–18].

Although malaria vectors still bite predominantly indoors at night [19], a systematic review of human-vector interactions across sub-Saharan Africa (SSA) estimated that the proportion of bites occurring outdoors had risen by 10% between 2003 and 2018 [20]. This increase in the proportion of outdoor biting has been predicted to result in an additional 10.6 million malaria cases a year in SSA assuming universal coverage with ITNs and IRS is achieved [20]. These findings highlight the need to characterise the risk of malaria transmission in the context of the increasing use of interventions.

A standard measure of malaria transmission is the entomological inoculation rate (EIR), the mean number of infective bites per person per unit of time. The EIR is estimated by multiplying the biting rate, estimated from the number of host-seeking mosquitoes caught, by the estimated proportion of mosquitoes that are infected with sporozoites [21]. The EIR is useful for quantifying the risk of infective bites. However, it does not capture the actual risk experienced by the community since it does not include the use of personal protective interventions. The component metrics for EIR are typically obtained over the entire active mosquito-biting period, usually all night, and do not take changes in biting rates, the proportion of infected-bites and human behaviour throughout the night into account.

Estimating the risk for individual hours throughout the night may allow a more accurate estimation of the community's actual risk of malaria. The locations of humans (whether inside or outside dwellings and whether under a net) throughout the night are needed to properly characterise the actual risk of malaria infections [22,23]. The relevance of *Anopheles* bites measured by catches of host-seeking mosquitoes depends on the availability of unprotected humans at the respective times and locations [22,24,25]. Some studies have also indicated that mosquitoes with *Plasmodium* sporozoites or those that were parous and therefore more likely to be infected may bite at different times of the night (S1 Table).

Malaria risk metrics that are more granular and that take human behaviour into account may improve the identification of gaps in the existing protection and assist the designing of

more effective intervention responses. There are several studies that overlap human behaviour and entomological data [19,26–44], some of which have hourly human behaviour data with host-seeking mosquitoes and mosquito infection data in the presence of high ITN coverage. Among recent studies in mainland Africa, a report from Burkina Faso combined entomological and human behaviour data and found that while the majority of infective bites would have occurred during times when a substantial proportion of people were using ITN, transmission outside these hours still occurred [30]. One study in Mozambique found a similar pattern with host-seeking mosquitoes [31], while another reported a higher outdoor exposure [26]. A study in Western Kenya found that while ITN use protected against the majority of bites there was a risk of biting in the early morning [28].

A previous study in 2016–17 in Tanzania linked human behaviour data to mosquito densities primarily caught using CDC light traps [33]. In the present study we use the same Tanzanian human behaviour data but link it to a slightly later entomological study in the same area with hourly data on infective bites from the number of indoor and outdoor host-seeking mosquitoes caught by human landing catch and the proportion of infected mosquitoes. The present study aims to understand the patterns of human exposure to malaria-infected mosquitoes in rural Tanzanian villages where ITNs and IRS are used.

## Methods

### Study area

This study was conducted in Ulanga and Kilombero districts in south-eastern Tanzania (Fig 1) in the greater Kilombero valley. The climate is mostly hot and humid. The annual rainfall is 1200-1800mm, with a peak between October and November and a second peak between April and May, while temperatures range between 20˚C and 33˚C [45]. The communities practise rice farming in irrigated paddies, subsistence agriculture and small-scale fishing. Recent

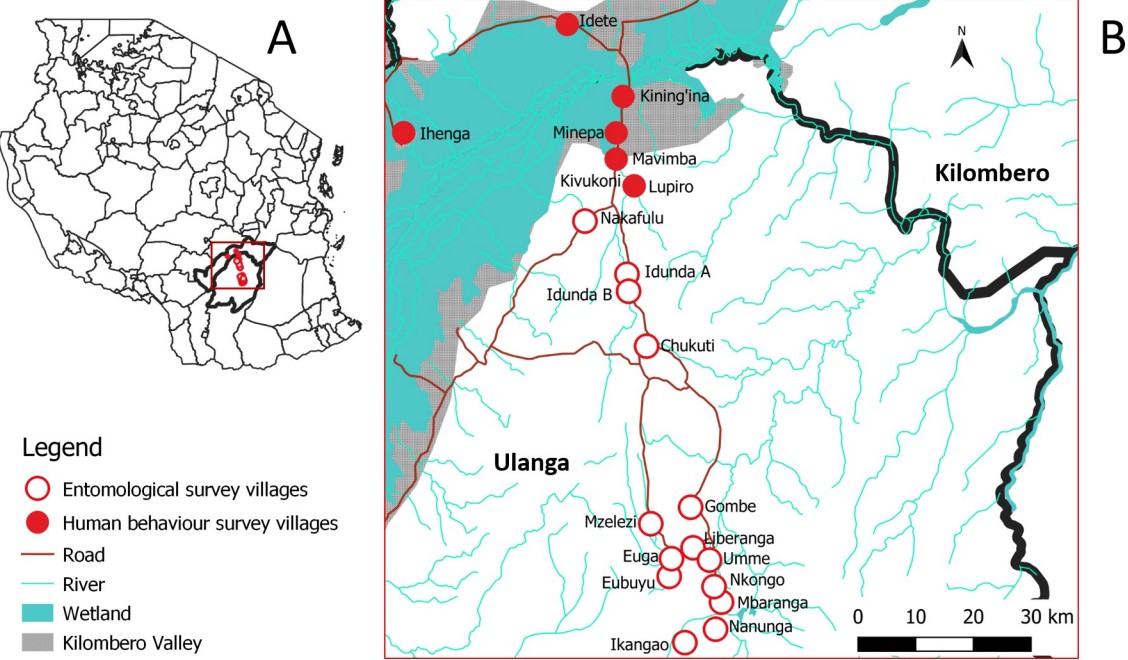

**Fig 1. Map of study area.** (**A**) Location of Kilombero and Ulanga Districts (bold line borders) in Tanzania. (**B**) Locations of the study villages in Ulanga and Kilombero Districts. The map was created in Q-GIS using a shapefile from: https://diva-gis.org/.

studies have shown that *An. gambiae* sensu lato in this area consists almost entirely of *An. arabiensis*, while *An. funestus* s.l. comprises more than 95% *An. funestus sensu stricto* [46–50]. Moderate to high levels of malaria transmission occur all year with seasonal peaks around the wet seasons. ITNs are the primary vector control intervention in the area [50–52], and a mass campaign shortly before the human behaviour survey resulted in self-reported use of ITNs of 99%. In addition, a community-wide IRS with pirimiphos-methyl (Actellic 300, capsule suspension) and a perlite-mineral insecticide (Imergard, wettable powder) was also implemented between January and October in 2019 as part of an intervention trial [53]. At the time of the entomological survey, 80% of households had IRS.

## Study design

This study combines secondary analysis of data from two different studies: a human behaviour survey and an entomological survey.

**Human behaviour survey.** The human behaviour survey was conducted between August 2016 and November 2017 in rural communities comprising six villages (Kivukoni, Minepa, Lupiro, Idete, Ihenga and Kining'ina) in Ulanga and Kilombero districts and in urban and peri-urban settlements comprising three sites (Katindiuka, Ifakara Mjini and Viwanja Sitini) in Kilombero district. The surveys were done around and inside houses from dusk to dawn, and have been described in detail elsewhere [33]. Ninety households from the villages (ten houses per village) were randomly selected from a house enumeration list from the Ifakara Health and Demographic Surveillance System [54]. Consenting adult household members were recruited and provided with three-day training on how to observe and record nightly household activities among members of their households. The number of people doing different activities at different times and locations generally classified as outdoors, indoors but out of bed and in bed with or without ITN use were recorded every half hour from 6PM to 6AM. The person's location was only recorded at the half-hour time-point and not in between time-points. The household members were classified by sex and age (adults and children of school-going age (aged 6 years and above) and children below school-going age (under 6 years)). The observations were made for three days every month, for three months in the rainy season and another three months in the dry season. For purposes of this current study, data from peri-urban and urban settlements as captured in the original study [33] have been excluded to match the entomological surveys which were all done in rural communities.

**Entomological survey.** The entomological survey was conducted between August 2018 and September 2019 in fourteen villages in Ulanga district (Nakafulu, Idunda A, Idunda B, Chikuti, Gombe, Liberanga, Umme, Nkongo, Mbaranga, Ikangao, Eubuyu, Euga, Mzelezi and Nanunga) (Fig 1). The villages had been deliberately selected as having substantial mosquito populations for the purpose of the IRS evaluation. Host-seeking mosquitoes were collected by human landing catches (HLC) between 6PM to 6AM by a pair of volunteers alternating positions indoors and outdoors every hour [55]. The collections were conducted in three houses selected from each of fourteen villages and were repeated for six nights per month over eight months. Similar to the human behaviour survey, the entomological survey included the wet and dry seasons. The entomological survey has been described in detail elsewhere [56]. Female *Anopheles* mosquitoes were identified morphologically. Polymerase chain reaction (PCR) assays were conducted to identify the sibling species of the *An. gambiae* and *An. funestus* complexes [57,58]. *Plasmodium* circumsporozoite protein (CSP) tests were done by enzyme-linked immunosorbent assays (ELISA) to detect mosquitoes infected with malaria parasites [59].

## Data analysis

We adapted the notation and formulae from initial work by Monroe *et al* for measuring human exposure to bites occurring in different locations and whether protected or unprotected [60].

**Total and infected mosquito biting rates.** The biting rates, the mean number of mosquito bites per person, were estimated using the HLC collections. For each hour, *t*, and species, *m*, the outdoor, $B_{O,t,m}$, and indoor, $B_{I,t,m}$, biting rates were estimated using Poisson regression with crossed random effects to account for repeated observations by household and date and a fixed effect for hour. This takes into account the unbalanced sampling by household, date and hour. Separate models were run for each species and location.

The proportions of mosquitoes infected with sporozoites, $p_{l,t,m}$, were estimated with exact binomial confidence intervals for each location, *l*, hour, *t*, and species, *m*. Due to low numbers of infected mosquitoes, clustering was not accounted for. We also aggregated the results for the proportions of infected mosquitoes for some hours when calculating the rates of infective bites: for indoors, the categories were 6PM-11PM, 11PM-12AM, 12AM-1AM, 1AM-6AM and for outdoors, all hours were pooled together.

The mean number of infective bites per person outdoors, and indoors, were estimated per hour by multiplying the hourly biting rates by the proportion of bites from the infected mosquitoes.

A sum of the hourly *An. arabiensis* and *An. funestus* infective bites was then obtained to estimate the total hourly incidence of infective bites. We estimated the 95% confidence intervals for the infective biting rates taking the uncertainty for both the overall biting rates and the proportion of mosquitoes infected into account. For each hour, species and location, we randomly sampled 1000 draws from the distribution for the number of bites per person per hour (we used a normal distribution parameterized with our estimated mean biting rate and standard error (SE)) and from the distribution of the proportion of mosquitoes infected (we used a normal distribution with our estimated mean proportion infected and SE). We calculated the number of infective bites for each of the 1000 samples. The 2.5 and 97.5 percentiles of the 1000 samples yielded the 95% confidence interval. We assume that the covariance between biting rates and the proportion of mosquitoes infected is zero.

We also aggregated the outdoor and indoor hourly infective bites over three time intervals. These intervals represent times when mostly people are outdoors or indoors prior to bed time (6AM to 10PM), during bed time (10PM to 5AM) and after bed time (later than 5AM) where different interventions would need to be used.

**Human exposure to infected mosquito bites in the absence of ITNs.** For each hour of the night, *t*, between 6PM and 6AM, the proportion of recorded times that people were outdoors, $O_t$, indoors in bed (during sleep), $S_t$, with or without ITNs, and indoors out of bed (awake), $A_t$, were estimated, using robust standard errors to account for clustering by household. The human behaviour survey recorded the location of participants indoors as 'in bed' or 'out of bed' while the Monroe *et al* notation classifies participants as 'sleeping' or 'awake'. For purposes of using this notation, we assume that these are synonymous. The mean number of infective bites indoors awake per person per night, $n_A$ was estimated by the sum for all the hours and species of the time at risk multiplied by the incidence of infective bites, so that $n_A = \sum_m \sum_t B_{I,t,m} p_{I,t,m} A_t$. Similarly, the mean number of infective bites indoors while in bed sleeping per person per night assuming no net use, was given by $n_{S,u} = \sum_m \sum_t B_{I,t,m} p_{I,t,m} S_t$, and the mean number of infective bites per person per night outdoors by $n_O = \sum_m \sum_t B_{O,t,m} p_{O,t,m} O_t$. The proportion of infective bites estimated to occur indoors and in bed sleeping assuming no net use was given by $\pi_{S,u} = \frac{n_{S,u}}{n_A + n_{S,u} + n_O}$.

**Proportion of infective bites occurring when people are using ITNs.** For each hour of the night, $t$, the proportion of time spent in bed and protected by ITNs, $S_{p,t}$ was estimated. The proportion of infective bites occurring when people were using ITNs, $P_S^*$, was estimated by the sum over the hours $t$ of infective bites occurring during sleep multiplied by the proportion of time in hour $t$ that ITNs were used while sleeping divided by the number of all infective bites. This would represent the proportion of infective bites averted by ITNs if ITNs prevent 100% of bites while in use. Normally, ITNs do not block every single mosquito bite while in use [61–63], and so $P_S^*$ would represent the maximum protection for ITN users in this study setting. This potential maximum protection from ITNs was estimated separately for children below school age and the rest of the household members.

**Estimated number of infective bites per person per year.** We summed the estimated infective bites over the night (taking the mean of indoor and outdoor bites) to give an estimate of the EIR without incorporating human behaviour. We calculated a similar measure taking into account human location and ITN use to estimate the actual transmission risk experienced by the community.

Data analysis was carried out using Stata (16.1, StataCorp LLC, College Station, TX) and R version 4.3.2 [64].

Table 1 describes the quantities in the model.

## Ethics approval and consent to participate

Ethical clearance was obtained from Ifakara Health Institute Review Board, (Entomological surveys: IHI IRB 021/2016 & 015/2017, Human behaviour surveys: IHI/IRB/No: IHI/IRB/No: 35–2015) and the Medical Research Coordinating Committee of the Tanzanian National Institute of Medical Research (Entomological surveys: NIMR/HQ/R.8a/Vol.IX/1725 & 2270, Human behaviour surveys: NIMR/HQ/R.8a/Vol.IX/2162). Written informed consent was obtained from all household heads and volunteers prior to their participation in the surveys. Written informed consent was obtained from the parent/guardian of each participant under 18 years of age. Permission to publish this work was granted by Tanzanian National Institute of Medical Research: NIMR/HQ/P.12 VOL.XXXV/164.

**Table 1. Quantities used to estimate behaviour-adjusted exposure to *Anopheles* infective bites.**

| Quantity | Description |
| --- | --- |
| $l$ | Location (Indoors or Outdoors) |
| $t$ | Hour |
| $m$ | Species (*An. arabiensis* or *An. funestus*) |
| $B_{O,t,m}$ | Number of outdoor bites in hour $t$ by species $m$ |
| $B_{I,t,m}$ | Number of indoor bites in hour $t$ by species $m$ |
| $p_{l,t,m}$ | Proportion of bites that came from infective mosquitoes |
| $O_t$ | Proportion of time spent by humans outdoors in hour $t$ |
| $S_t$ | Proportion of time spent by humans indoors asleep in hour $t$ |
| $A_t$ | Proportion of time spent by humans indoors awake in hour $t$ |
| $n_A$ | Number of infective bites per person indoors awake per night |
| $n_O$ | Number of infective bites per person outdoors per night |
| $n_{S,u}$ | Number of infective bites per person indoors asleep per night assuming no net use |
| $\pi_{S,u}$ | Proportion of infective bites that occur indoors asleep assuming no net use |
| $S_{p,t}$ | Proportion of time spent by humans indoors asleep using a bed net in hour $t$ |
| $P_S^*$ | Proportion of infective bites occurring during sleep when protected by a net |

**Table 2. Mosquito biting rates estimated from HLC.**

| | Indoors | | | Outdoors | | |
|---|---|---|---|---|---|---|
| | Number of HLC | Estimated mean number of bites/person/hour (95% CIs) | | Number of HLC | Estimated mean number of bites/person/hour (95% CIs) | |
| | | *An. arabiensis* | *An. funestus* | | *An. arabiensis* | *An. funestus* |
| 6-7PM | 171 | 0.10 (0.06, 0.18) | 0.51 (0.36, 0.70) | 162 | 0.10 (0.05, 0.19) | 0.38 (0.27, 0.54) |
| 7-8PM | 208 | 0.11 (0.06, 0.19) | 0.98 (0.72, 1.33) | 207 | 0.12 (0.06, 0.23) | 0.86 (0.64, 1.16) |
| 8-9PM | 212 | 0.10 (0.06, 0.17) | 0.96 (0.71, 1.30) | 215 | 0.10 (0.05, 0.20) | 1.18 (0.88, 1.58) |
| 9-10PM | 213 | 0.11 (0.06, 0.19) | 1.05 (0.78, 1.43) | 215 | 0.12 (0.06, 0.22) | 1.23 (0.91, 1.65) |
| 10-11PM | 222 | 0.12 (0.07, 0.20) | 1.08 (0.80, 1.46) | 214 | 0.12(0.06, 0.23) | 1.16 (0.87, 1.57) |
| 11-12AM | 207 | 0.12 (0.07, 0.21) | 1.07 (0.79, 1.45) | 218 | 0.10 (0.05, 0.20) | 1.05 (0.78, 1.41) |
| 12-1AM | 205 | 0.12 (0.07, 0.21) | 0.95 (0.70, 1.29) | 202 | 0.08 (0.04, 0.16) | 1.12 (0.83, 1.50) |
| 1-2AM | 204 | 0.09 (0.05, 0.17) | 1.03 (0.76, 1.40) | 205 | 0.11 (0.06, 0.22) | 1.13 (0.84, 1.52) |
| 2-3AM | 197 | 0.13 (0.07, 0.22) | 1.04 (0.76, 1.41) | 206 | 0.11 (0.05, 0.21) | 1.09 (0.81, 1.47) |
| 3-4AM | 208 | 0.08 (0.04, 0.14) | 1.09 (0.81, 1.48) | 211 | 0.11 (0.06, 0.22) | 1.12 (0.83, 1.50) |
| 4-5AM | 197 | 0.07 (0.04, 0.12) | 1.08 (0.80, 1.47) | 195 | 0.07 (0.03, 0.13) | 1.21 (0.90, 1.62) |
| 5-6AM | 160 | 0.06 (0.03, 0.11) | 0.90 (0.66, 1.23) | 159 | 0.04 (0.02, 0.09) | 1.03 (0.76, 1.40) |

The mean number of bites per person per hour was estimated using Poisson regression with crossed random effects for house and date and a fixed effect for hour. Separate analyses were run for indoor and outdoor locations, and by species.

## Results

### Indoor and outdoor biting rates

There were HLC collections in 46 households from 14 villages over 62 dates. A total of 8,276 *An. funestus* s.s. and 1,927 *An. arabiensis* mosquitoes were caught. The biting rates were higher for *An. funestus* compared to *An. arabiensis*, but the split between indoor and outdoor biting was similar for the two species (Table 2 and Fig 2).

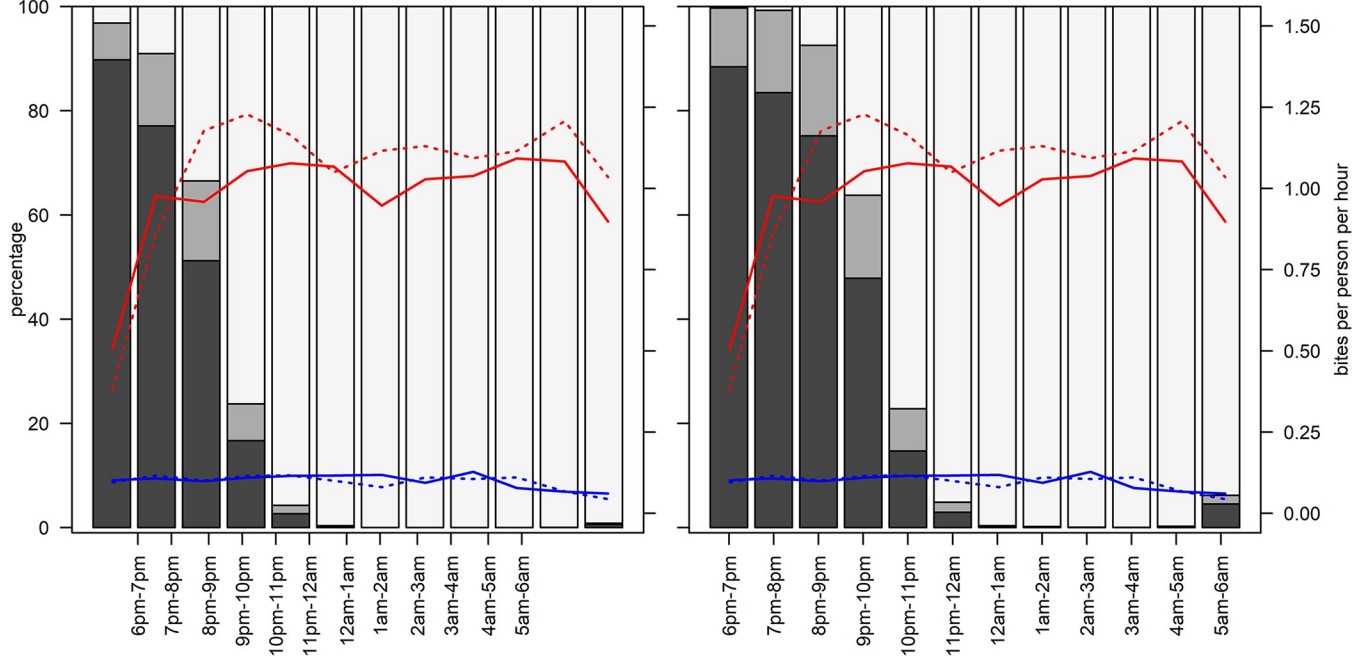

**Fig 2. Locations of household members and *Anopheles* bites at night.**

**Table 3. Proportion of mosquitoes infected.**

| | *An. arabiensis* | | | | *An. funestus* | | | |
|---|---|---|---|---|---|---|---|---|
| | Number tested | Number positive | Estimated proportion infected (95% CI[1]) | | Number tested | Number positive | Estimated proportion infected (95% CI[1]) | |
| *Indoors* | | | | | | | | |
| 6-7PM | 81 | 0 | 0 | (0, 0.04) | 365 | 2 | 0.005 | (0.001, 0.02) |
| 7-8PM | 67 | 0 | 0 | (0, 0.05) | 508 | 0 | 0 | (0, 0.007) |
| 8-9PM | 70 | 0 | 0 | (0, 0.05) | 496 | 1 | 0.002 | (0.0001, 0.01) |
| 9-10PM | 82 | 0 | 0 | (0, 0.04) | 413 | 1 | 0.002 | (0, 0.01) |
| 10-11PM | 1630 | 4 | 0.002 | (0.001, 0.006) | 17731 | 75 | 0.004 | (0.003, 0.005) |
| 11-12AM | 2006 | 4 | 0.002 | (0.001, 0.005) | 17424 | 82 | 0.005 | (0.004, 0.006) |
| 12-1AM | 1606 | 4 | 0.002 | (0.001, 0.006) | 16780 | 77 | 0.005 | (0.004, 0.006) |
| 1-2AM | 1711 | 5 | 0.003 | (0.001, 0.007) | 17065 | 80 | 0.005 | (0.004, 0.006) |
| 2-3AM | 77 | 0 | 0 | (0, 0.05) | 375 | 4 | 0.011 | (0.003, 0.027) |
| 3-4AM | 48 | 0 | 0 | (0, 0.07) | 405 | 2 | 0.005 | (0.001, 0.018) |
| 4-5AM | 39 | 0 | 0 | (0, 0.09) | 393 | 1 | 0.003 | (0.0001, 0.014) |
| 5-6AM | 25 | 0 | 0 | (0, 0.14) | 264 | 1 | 0.004 | (0.0001, 0.021) |
| *Outdoors* | | | | | | | | |
| 6-7PM | 82 | 0 | 0 | (0, 0.04) | 121 | 0 | 0 | (0, 0.030) |
| 7-8PM | 102 | 0 | 0 | (0, 0.04) | 272 | 1 | 0.004 | (0.0001, 0.020) |
| 8-9PM | 94 | 0 | 0 | (0, 0.04) | 375 | 0 | 0 | (0, 0.010) |
| 9-10PM | 110 | 0 | 0 | (0, 0.03) | 402 | 1 | 0.002 | (0.0001, 0.014) |
| 10-11PM | 108 | 0 | 0 | (0, 0.05) | 396 | 2 | 0.005 | (0.0001, 0.018) |
| 11-12AM | 98 | 0 | 0 | (0, 0.04) | 337 | 2 | 0.006 | (0.0001, 0.021) |
| 12-1AM | 77 | 0 | 0 | (0, 0.04) | 353 | 1 | 0.003 | (0.0001, 0.016) |
| 1-2AM | 98 | 0 | 0 | (0, 0.04) | 371 | 3 | 0.008 | (0.002, 0.023) |
| 2-3AM | 97 | 0 | 0 | (0, 0.04) | 346 | 0 | 0 | (0, 0.011) |
| 3-4AM | 91 | 0 | 0 | (0, 0.04) | 380 | 3 | 0.008 | (0.002, 0.023) |
| 4-5AM | 53 | 0 | 0 | (0, 0.07) | 384 | 0 | 0 | (0, 0.010) |
| 5-6AM | 34 | 0 | 0 | (0, 0.10) | 288 | 0 | 0 | (0, 0.013) |

[1]Exact binomial 95% confidence intervals.

### Proportion of mosquitoes that were infected with *Plasmodium falciparum*

The proportion of mosquitoes infected with *P. falciparum* sporozoites tended to be higher in mosquitoes collected indoors than in those collected outdoors (Table 3). For *An. funestus* s.s. the proportion infected indoors was 0.005 (326/72219) and outdoors was 0.003 (13/4025). For *An. arabiensis*, the proportion infected was 0.002 (17/7442) indoors and 0 (0/1044) outdoors. There were small numbers of infected mosquitoes and no clear patterns with time. The uncertainty, represented by the width of the CI, was greatest where few mosquitoes were available for testing due to low biting rates. For this reason, for the estimates of behaviour-adjusted exposure the proportion of mosquitoes infected were aggregated over multiple hours.

### Rate of infective bites

The estimated rate of infective bites varied by time of the night, species and location (Table 4). The estimated infective biting rates were slightly higher indoors compared to outdoors, and were lowest in the early evening and late morning.

**Table 4. Estimated rates of infective bites.**

| | Infective bites per 100 person-hours (95% CI) | | | | | |
|---|---|---|---|---|---|---|
| | | | | | | |
| | Indoors | | | Outdoors | | |
| | *An. arabiensis* | *An. funestus* | total | *An. arabiensis* | *An. funestus* | total |
| 6-7PM | 0.02 (0.01, 0.04) | 0.21 (0.14, 0.30) | 0.23 (0.16, 0.32) | 0 | 0.12 (0.05, 0.21) | 0.12 (0.05, 0.21) |
| 7-8PM | 0.02 (0.01, 0.04) | 0.40 (0.27, 0.57) | 0.42 (0.29, 0.60) | 0 | 0.28 (0.12, 0.48) | 0.28 (0.12, 0.48) |
| 8-9PM | 0.02 (0.01, 0.04) | 0.39 (0.25, 0.56) | 0.41 (0.27, 0.58) | 0 | 0.39 (0.17, 0.66) | 0.39 (0.17, 0.66) |
| 9-10PM | 0.02 (0.01, 0.04) | 0.44 (0.30, 0.63) | 0.46 (0.32, 0.65) | 0 | 0.40 (0.16, 0.66) | 0.40 (0.16, 0.66) |
| 10-11PM | 0.02 (0.01, 0.05) | 0.44 (0.29, 0.66) | 0.47 (0.31, 0.69) | 0 | 0.38 (0.17, 0.65) | 0.38 (0.17, 0.65) |
| 11-12AM | 0.02 (0.01, 0.05) | 0.51 (0.33, 0.73) | 0.53 (0.36, 0.76) | 0 | 0.34 (0.14, 0.59) | 0.34 (0.14, 0.59) |
| 12-1AM | 0.03 (0.01, 0.06) | 0.44 (0.30, 0.62) | 0.47 (0.33, 0.65) | 0 | 0.37 (0.16, 0.62) | 0.37 (0.16, 0.62) |
| 1-2AM | 0.03 (0.01, 0.05) | 0.49 (0.33, 0.70) | 0.52 (0.35, 0.73) | 0 | 0.37 (0.16, 0.63) | 0.37 (0.16, 0.63) |
| 2-3AM | 0.03 (0.02, 0.06) | 0.50 (0.33, 0.70) | 0.53 (0.36, 0.75) | 0 | 0.36 (0.15, 0.60) | 0.36 (0.15, 0.60) |
| 3-4AM | 0.02 (0.01, 0.04) | 0.52 (0.36, 0.74) | 0.54 (0.38, 0.77) | 0 | 0.36 (0.16, 0.62) | 0.36 (0.16, 0.62) |
| 4-5AM | 0.02 (0.01, 0.03) | 0.51 (0.34, 0.72) | 0.53 (0.36, 0.74) | 0 | 0.39 (0.16, 0.66) | 0.39 (0.16, 0.66) |
| 5-6AM | 0.02 (0.01, 0.03) | 0.43 (0.29, 0.61) | 0.45 (0.31, 0.63) | 0 | 0.34 (0.15, 0.57) | 0.34 (0.15, 0.57) |

The rate of infected bites was estimated by combining the biting rates and the proportion of mosquitoes infected (sporozoite rates). We aggregated the proportion of mosquitoes infected from 6-11PM, 11PM-12AM, 12-1AM, 1-6AM indoors and all hours outdoors due to small numbers.

The percentage of infective bites occurring between 10PM and 5AM (representing 58% of the 12 hour period of HLC) was estimated to be 65% (60%, 70%) indoors and 63% (54%, 71%) outdoors.

## Observations of human behaviours and activities indoors and outdoors

Sixty households had records spread over three months, with a median of 8 nights per household with range 2 to 18.

There was a total of 170,257 observations of the locations of individuals made at half-hourly intervals. Overall, the majority of the observations of participants in the early evenings between 6PM and 9PM were outdoors (Table 5). Between 9PM and 10PM, the proportion of time spent outdoors dropped and by midnight, nearly all observations recorded were of individuals indoors. The proportion of recorded locations of individuals that were indoors in bed rose steadily each hour from 9PM to midnight. Nearly everyone who was recorded at 6AM, was still indoors in bed. Time spent in bed tended to be longer for children below school age than for older household members (Fig 3).

## Proportion of infective bites during times spent under ITNs

The proportion of time spent under ITNs out of the total time spent in bed, was high in both children below school age, 99.2% (95% CI 97.0%, 99.8%) and older household members, 98.8% (95% CI 97.2%, 99.5%). Nearly everyone who was recorded at midnight onwards was indoors in bed and under an ITN (Fig 3).

The proportion of infective bites between 6PM and 6AM that occurred during times when the individuals were sleeping under ITNs was estimated to be 85% (81%, 88%) for children below school age and 76% (71%, 81%) for older household members. The percentage of infective bites that would occur when people were outdoors was estimated to be 11% (8%, 15%) for children below school age children and 17% (13%, 22%) for older participants (Fig 4).

**Table 5. Proportion of people observed by location.**

| | Children below school-age | | | School-age children and adults | | |
|---|---|---|---|---|---|---|
| | Outdoors %(95% CIs) | Indoors out of bed % (95% CIs) | Indoors in bed % (95% CIs) | Outdoors %(95% CIs) | Indoors out of bed % (95% CIs) | Indoors in bed % (95% CIs) |
| 6-7PM | 89.8 (84.0, 93.6) | 7.0 (3.7, 13.1) | 3.2 (1.7, 6.0) | 88.4 (84.4, 91.5) | 11.3 (8.2, 15.2) | 0.3 (0.1, 0.6) |
| 7-8PM | 77.1 (70.3, 82.7) | 13.9 (8.7, 21.3) | 9.1 (6.1, 13.2) | 83.4 (78.9, 87.1) | 15.8 (12.1, 20.3) | 0.7 (0.4, 1.2) |
| 8-9PM | 51.2 (44.7, 57.7) | 15.3 (8.8, 25.3) | 33.5 (27.1, 40.5) | 75.1 (69.5, 80.1) | 17.4 (13.0, 22.9) | 7.5 (5.5,10.1) |
| 9-10PM | 16.7 (12.3, 22.2) | 7.0 (3.1, 15.3) | 76.3 (68.5, 82.6) | 47.9 (41.8, 53.9) | 15.9 (11.9, 21.0) | 36.2 (30.1, 42.8) |
| 10-11PM | 2.7 (1.6, 4.4) | 1.6 (0.8, 3.2) | 95.7 (93.2, 97.4) | 14.7 (10.8, 19.7) | 8.1 (5.1, 12.6) | 77.2 (71.1, 82.3) |
| 11-12AM | 0.1 (0.03, 0.6) | 0.2 (0.1, 0.6) | 99.6 (99.2, 99.8) | 2.9 (1.5, 5.8) | 1.9 (1.2, 3.2) | 95.1 (92.1, 97.0) |
| 12-1AM | 0 | 0 | 100 | 0.2 (0.04, 0.6) | 0.2 (0.03, 1.2) | 99.6 (98.8,99.9) |
| 1-2AM | 0 | 0 | 100 | 0 | 0.2 (0.06, 0.7) | 99.8 (99.3, 99.9) |
| 2-3AM | 0 | 0 | 100 | 0 | 0.007 (0, 0.2) | 99.9 (99.8, 100) |
| 3-4AM | 0 | 0 | 100 | 0 | 0.04 (0, 0.2) | 100 (99.8, 100) |
| 4-5AM | 0.03 (0, 0.2) | 0 | 99.9 (99.8, 100) | 0.2 (0.07, 0.5) | 0.06 (0, 0.1) | 99.7 (99.4, 99.9) |
| 5-6AM | 0.6 (0.02, 1.7) | 0.2 (0.04, 1.3) | 99.2 (97.7, 99.7) | 4.5 (3.1, 6.6) | 1.7 (1.0, 2.7) | 93.8 (91.4, 95.6) |

Percentages of time spent in different locations were estimated as the proportion of half hours spent in the locations per hour out of the total half hours spent by the population in the respective hour. Robust standard errors were used to account for repeated observations by household.

We summed the estimated infective bites for each hour averaging the indoor and outdoor collections to give an estimate of the EIR without incorporating human behaviour. For the months of the entomological collections, the standard EIR was estimated to be equivalent to

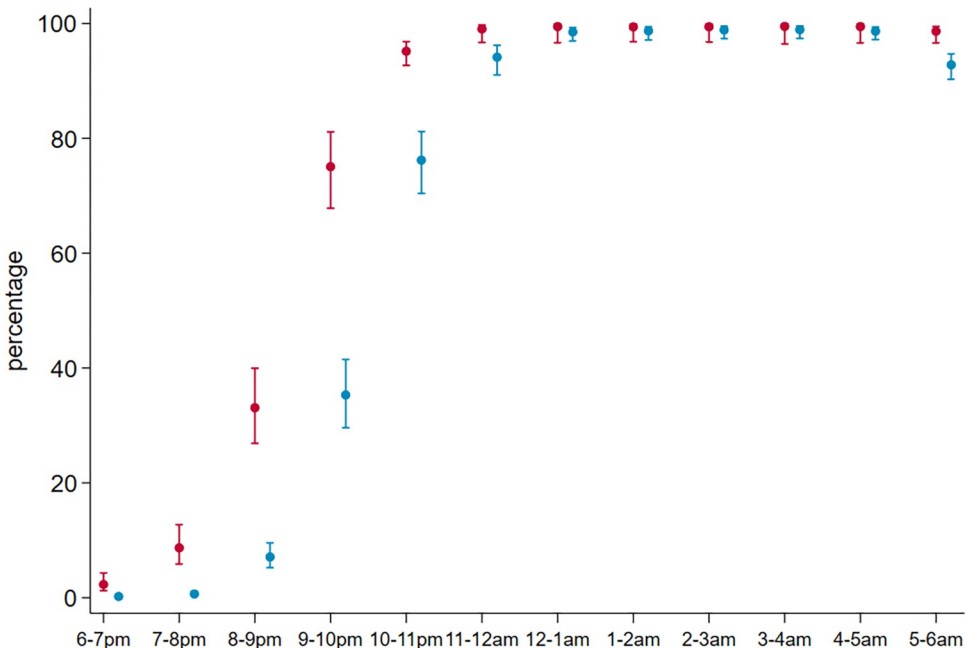

**Fig 3. Hourly use of ITNs in the household.**

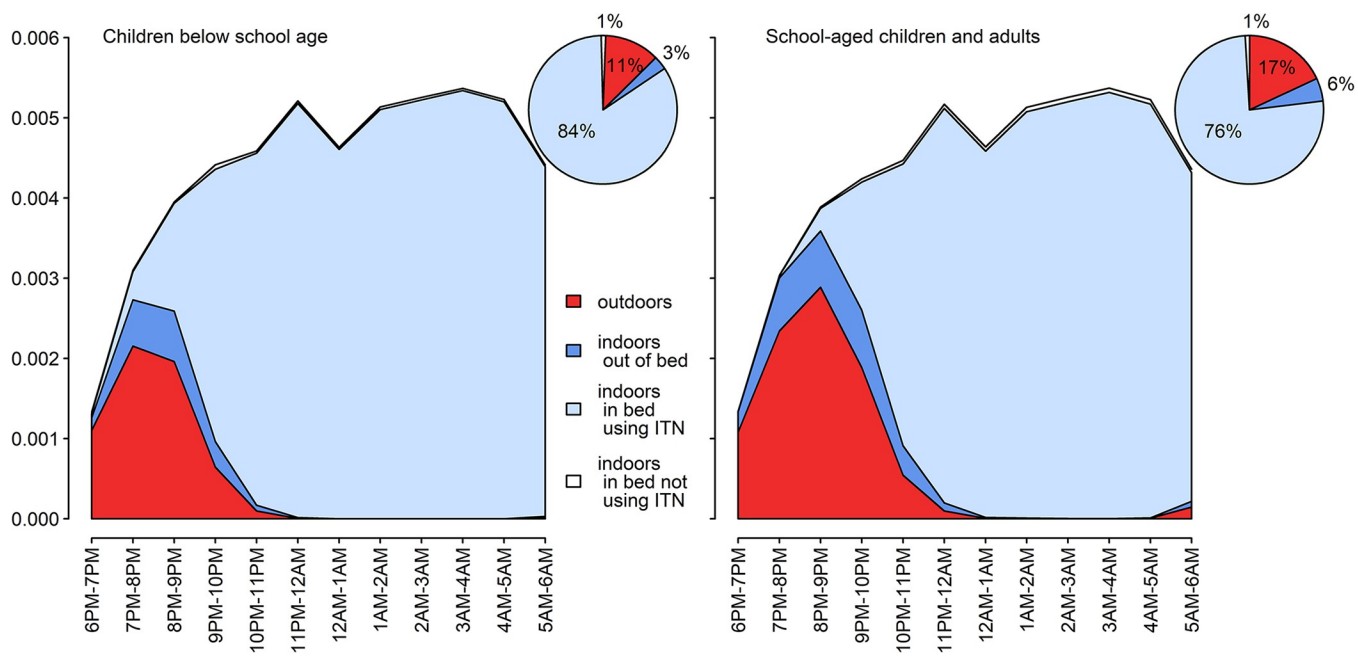

**Fig 4. Human exposure to malaria and use of ITNs across the night.**

17.7 infective bites per person per year. Taking human location and ITN use into account (and assuming 100% protection from ITN), the mean rate of infective bites was estimated to be 4.6 per year for older participants and 3.0 for children below school-age.

## Discussion

Residual malaria transmission has been raised as a potential challenge for malaria control programmes [35,65–67] and can be in part due to outdoor biting as well as other factors [68]. In order to address gaps in malaria vector control, it is necessary to understand the behaviours of mosquito vectors and of human hosts that together can result in exposure to infective mosquito bites. In a setting with high ITN coverage and recent application of IRS, we investigated where and at what time during the night, *Plasmodium* sporozoite-positive local malaria vectors bite human hosts, and quantified the proportion of infective bites that would occur when household members are using ITNs in this setting.

While both species contribute to transmission in this area [33,46,69], *An funestus* s.s contributed higher numbers of mosquitoes caught and higher proportions of sporozoite-positive mosquitoes compared to *An. arabiensis* consistent with the recent finding of the relatively higher importance of *An. funestus* s.s for malaria transmission in the area [70]. *An. arabiensis* has previously been thought to be the main agent of outdoor malaria transmission owing to reports of predominant outdoor biting tendencies [15,71,72]. However, in our study, none of the *An. arabiensis* mosquitoes caught outdoors were infected, and other studies in the same area have also reported a generally low proportions of infection [46,70]. These findings may suggest a limited role for outdoor biting by *An. arabiensis* in malaria transmission in the area.

There was variation in biting rates and the proportion of mosquitoes infected. The biting rates were similar indoors and outdoors, but varied by time being lowest in the early evening and after 5AM. The proportions of mosquitoes infected tended to be higher for mosquitoes caught indoors compared to outdoors, and between 10PM and 3AM compared to other times

during the night. Previous studies have reported variations in both biting rates and the proportions of sporozoite-positive or parous mosquitoes at different times of the night (S1 Table), although the differences are not consistent.

In our study, an estimated 65% (95% CI 60%, 70%) of the indoor infective bites occurred between 10PM and 5AM. A separate study [73] in the same region estimated that 8% of the infected *An gambiae* caught by light traps were between 7PM and 10PM, at hours of the night when people were unlikely to use a mosquito net. The very high reported ITN use in our study: 99.2% (95% CI 97.0%, 99.8%) among children below school age and 98.8% (95% CI 97.2%, 99.5%) among older household members in this study could potentially protect these groups against 85% (95% CI 81%, 88%) and 76% (71%, 81%) of infective bites that they would be exposed to between 6PM and 6AM assuming 100% protection (Fig 4).

We assumed that 100% of bites are prevented while under a net. We recognize that this is untrue and represents an upper bound: it is likely that protection starts off reasonably high but less than 100% and wanes as the net ages. Nets have been found to still offer some protection when there is insecticide resistance [74] or when old and with holes [61]. A modelling study would be able to predict the proportion of bites prevented over the lifespan of different types of ITNs, taking human behaviour into account. We also assume that the biting rates in the entomological study would be the same for all age-groups in the human behaviour study. There is evidence of different biting rates by host size [75,76], and that residents carrying out activities other than HLC may affect mosquito landing. This would not affect comparisons within the same age-group but would affect comparisons across age-groups and absolute levels of risk.

A disadvantage to characterizing risk during the night is the need for sufficient data to characterize each segment of the night. There were only 356 infected mosquitoes in the study. This led to imprecision for some hours and locations for the estimated proportions of mosquitoes infected, and consequently we aggregated across some time-periods. A larger study size would be beneficial, however since the aggregation focused on time periods when there were relatively fewer mosquitoes biting, it is unlikely to substantially alter the estimated peak period for infective bites during the night. We also do not capture day biting, which has been reported to contribute to transmission in a study in the Central African Republic [77], or early morning biting after 6am. We also cannot capture exposure for people who were away from the household [33]. Another limitation was that the two datasets for the entomological and human behaviour data were collected at different times, and from different villages, even though all were in the Kilombero valley. Villages where entomological surveys were carried out were at higher altitudes (range 311 to 1884m above sea level) than villages where human surveys were done (255m to 298m) [78]. The human behaviour data was collected between August 2016 and November 2017, and the entomological data between August 2018 and September 2019. If the human and mosquito behaviour changed over the altitude range or between the years, then the estimates combining the human and mosquito data would not represent any specific location and time period accurately, and generalization would also be limited. We needed to account for the variance structure introduced by the cluster sampling in our study, and different analysis methods may lead to different estimates. We used random effects for household and night for the entomological data because our data was unbalanced and our question focused on comparing the biting rates between the hours. We used robust standard errors for the human behaviour data. We investigated alternative methods as a sensitivity analysis: assuming zero for the proportion of mosquitoes infected in the hours with few mosquitoes caught and using robust variance estimates for the entomological data both lead to slightly higher estimates for the proportion of infective bites between 10PM and 5AM.

The high ITN and IRS use in the study area may limit the generalizability of the study findings to areas with lower ITN and IRS coverage since these interventions may impact mosquito

biting locations and timing, and potentially also human behaviour. The high ITN use in our study area may have occurred for several reasons. It was self-reported, the human behaviour study occurred within two years of a mass distribution campaign and the communities are within an area where several malaria transmission control studies had been carried out and the community may be adequately sensitised about the benefits of nets. This may not apply to the villages where the entomological surveys were carried out. In addition, the human behaviour survey only captures people who are at home. Those who have gone out may be less likely to use a net.

Evidence from our study reaffirms the need for an intervention which protects people indoors when they are asleep, such as ITNs. Sustaining high levels of ITN use by ensuring sufficient availability within households and regular use of the nets at night by all household members remains key to reducing malaria transmission. Increased advocacy and community engagement to encourage the maintenance of ITNs [79] and increase their longevity [80,81] can contribute to higher use where population access to ITNs is suboptimal [82]. The overall effects of widespread ITN use extend beyond the direct protection offered to users, by diminishing the overall mosquito population [36,83–86]. In addition to IRS, indoor interventions and personal protection measures such as repellents may provide protection when individuals are not in bed [87–89].

While in this area, the majority of infective bites could be prevented by the use of mosquito nets while sleeping, a small proportion of the infective bites occurred outdoors before people retired to bed. Outdoor biting needs to be addressed and could be reduced directly by tools designed for outdoor biting. There is also some evidence that mosquitoes biting outdoors go indoors at least once during their life and can be impacted by indoor interventions [83,90].

## Conclusion

In the study area, a substantial proportion of infective bites occurred indoors between 10PM and 5AM. Maintaining high levels of access and use of ITNs remains an important means to reduce malaria transmission in this area. This study also contributes to the evidence of different biting rates and proportions of biting mosquitoes that are infective at different times and locations in the night. The findings have implications for estimating the actual risk of malaria transmission in a community.

## Supporting information

**S1 Table. Proportions of *Anopheles* mosquitoes caught and infected and parous bites early, middle and late-night estimated from different areas.**
(DOCX)

## Acknowledgments

We would like to thank the community volunteers for their generous offer to assist with the activities of our surveys and the Ulanga and Kilombero residents for agreeing to accommodate the activities of our studies in and around their homes. This paper is dedicated to the memory of our colleague Godfrey Ligema.

## Author Contributions

**Conceptualization:** Isaac Haggai Namango, Sarah J. Moore, Manuel W. Hetzel, Amanda Ross.

**Data curation:** Isaac Haggai Namango, Sarah J. Moore.

**Formal analysis:** Isaac Haggai Namango, Sarah J. Moore, Amanda Ross.

**Investigation:** Noely Makungwa, Alex J. Limwagu, Salum Mapua, Olukayode G. Odufuwa, Godfrey Ligema, Hassan Ngonyani, Isaya Matanila, Jameel Bharmal, Jason Moore, Marceline Finda, Fredros Okumu.

**Supervision:** Sarah J. Moore, Manuel W. Hetzel, Amanda Ross.

**Writing – original draft:** Isaac Haggai Namango, Sarah J. Moore.

**Writing – review & editing:** Isaac Haggai Namango, Sarah J. Moore, Carly Marshall, Adam Saddler, David Kaftan, Frank Chelestino Tenywa, Olukayode G. Odufuwa, Jameel Bharmal, Fredros Okumu, Manuel W. Hetzel, Amanda Ross.

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
