## [Decision Letter · Decision Letter 0]

29 Aug 2024

PGPH-D-23-02284

A matter of timing: Biting by malaria-infected Anopheles mosquitoes and the use of interventions during the night in rural south-eastern Tanzania

Dear Dr. Namango,

Thank you for submitting your manuscript entitled "A matter of timing: Biting by malaria-infected Anopheles mosquitoes and the use of interventions during the night in rural south-eastern Tanzania" to PLOS Global Public Health. It has now been reviewed by two referees and while the manuscript shows promise, it will require major revisions before it can be considered for publication.

The reviewers have provided detailed feedback (appended below) that highlights areas where the manuscript could be strengthened, including but not limited to the overestimation of exposure risk, overestimation of ITN protection, as well as inconsistency in and limited data collection timeframe. Please thoroughly address these along with all of the other points raised by the reviewers and submit your revised manuscript along with a detailed point-by-point response to the reviewers' comments.

Thank you once again for submitting your work to PLOS Global Public Health, we look forward to receiving your revised manuscript.

Kind regards,

Nsa Dada, MSc, PhD

Academic Editor

Reviewers' comments:

1. We do not publish any copyright or trademark symbols that usually accompany proprietary names, eg (R), (C), or TM  (e.g. next to drug or reagent names). Please remove all instances of trademark/copyright symbols throughout the text, including ® on page 6.

2. Figure 1: please (a) provide a direct link to the base layer of the map (i.e., the country or region border shape) and ensure this is also included in the figure legend; and (b) provide a link to the terms of use / license information for the base layer image or shapefile. We cannot publish proprietary or copyrighted maps (e.g. Google Maps, Mapquest) and the terms of use for your map base layer must be compatible with our CC-BY 4.0 license. 

Reviewer's Responses to Questions

**Comments to the Author**

1. Does this manuscript meet PLOS Global Public Health’s publication criteria? Is the manuscript technically sound, and do the data support the conclusions? The manuscript must describe methodologically and ethically rigorous research with conclusions that are appropriately drawn based on the data presented.

Reviewer #1: Yes

Reviewer #2: Yes

2. Has the statistical analysis been performed appropriately and rigorously?

Reviewer #1: Yes

Reviewer #2: Yes

3. Have the authors made all data underlying the findings in their manuscript fully available (please refer to the Data Availability Statement at the start of the manuscript PDF file)?

Reviewer #1: Yes

Reviewer #2: Yes

4. Is the manuscript presented in an intelligible fashion and written in standard English?

Reviewer #1: Yes

Reviewer #2: Yes

5. Review Comments to the Author

Reviewer #1: This study combines entomological and human behavior data to assess malaria transmission risk in rural Tanzania. It found that most infective mosquito bites occur indoors at night when people are using insecticide-treated nets (ITNs). High ITN use potentially prevents 76-85% of infective bites, highlighting the importance of maintaining effective ITN coverage.

Here are the key issues the authors should consider addressing in this manuscript:

1. Consistency in data collection timeframes: The entomological data and human behavior data were collected at different times (2019 vs 2016-2017) and in different villages within the same region. This temporal and spatial mismatch should be more explicitly acknowledged as a limitation.

2. While I appreciate the difficulty in attaining infected mosquitoes, only 356 infected mosquitoes were found, leading to imprecision in estimates for some hours and locations. The authors aggregated across time periods to address this, but should discuss this limitation more thoroughly.

3. Assumptions in analysis:

- The authors assume 100% protection from ITNs while in use, which is likely an overestimate. This assumption should be more clearly stated and its implications discussed.

- The analysis assumes biting rates are the same across all age groups, which may not be accurate. This limitation should be acknowledged.

4. Limited time frame: The study only covers 6PM to 6AM, potentially missing daytime or early morning biting. This limitation should be noted.

5. Self-reported ITN use: The very high reported ITN use (99%) may be influenced by self-reporting bias or the recent distribution campaign. This potential bias should be discussed.

6. Generalizability: The study was conducted in an area with recent IRS application and high ITN coverage. The authors should discuss how this might limit generalizability to other settings.

7. Statistical methods: The authors used different methods to account for clustering in the entomological and human behavior data. They should justify this choice and possibly conduct sensitivity analyses with alternative methods.

8. Outdoor exposure: While the study focuses on indoor biting, more discussion on the implications of the 11-17% outdoor biting could be valuable, especially in the context of residual transmission.

9. Practical implications: The discussion could be strengthened by more explicitly addressing the practical implications of these findings for malaria control programs.

10. Data visualization: Some of the figures, particularly Figure 2, are quite complex and could potentially be simplified or split into multiple figures for clarity.

Addressing these issues would strengthen the manuscript and provide a more comprehensive and nuanced presentation of the study's findings and limitations.

Reviewer #2: In general, the MS addresses a very important topic on time and place of risk of exposures to infectious malaria mosquitoes in the communities. This is important for reinforcing use of interventions in the communities and understanding the gaps in the interventions and when and how we can supplement them primary ones (ITNs). The MS is well written, analysis well captured, and limitations of the study well outlined given that this is a secondary analysis, and a study conducted about 8 years ago. One limitation though with most human behavior observations including this study that is not mentioned is that the hourly capture of spatial presence of individuals indoors and outdoors sometimes can overestimate exposure risk for location (indoor or outdoor). This is because an individual can be counted twice or several times in a single period of observation i.e. in that 30-minute observation in this study, an individual can move between outdoors/indoors several times especially in rural settings where housing structures do not contain all the amenities indoors and the way the communities interact and live in these societies. So, I think this should be included as a limitation in this study as well. Otherwise well done team.

6. PLOS authors have the option to publish the peer review history of their article (what does this mean?). If published, this will include your full peer review and any attached files.

**Do you want your identity to be public for this peer review?** For information about this choice, including consent withdrawal, please see our Privacy Policy.

Reviewer #1: **Yes: **Eric Ochomo

Reviewer #2: No

---

## [Decision Letter · Decision Letter 1]

5 Nov 2024

A matter of timing: Biting by malaria-infected Anopheles mosquitoes and the use of interventions during the night in rural south-eastern Tanzania

PGPH-D-23-02284R1

Dear Dr Namango,

We are pleased to inform you that your manuscript 'A matter of timing: Biting by malaria-infected Anopheles mosquitoes and the use of interventions during the night in rural south-eastern Tanzania' has been provisionally accepted for publication in PLOS Global Public Health.

Thank you again for supporting Open Access publishing; we are looking forward to publishing your work in PLOS Global Public Health. Congratulations again!

Sincerely,

Amy Kristine Bei

Academic Editor

Reviewer Comments (if any, and for reference):

Reviewer's Responses to Questions

**Comments to the Author**

1. If the authors have adequately addressed your comments raised in a previous round of review and you feel that this manuscript is now acceptable for publication, you may indicate that here to bypass the “Comments to the Author” section, enter your conflict of interest statement in the “Confidential to Editor” section, and submit your "Accept" recommendation.

Reviewer #1: All comments have been addressed

Reviewer #2: All comments have been addressed

2. Does this manuscript meet PLOS Global Public Health’s publication criteria? Is the manuscript technically sound, and do the data support the conclusions? The manuscript must describe methodologically and ethically rigorous research with conclusions that are appropriately drawn based on the data presented.

Reviewer #1: Yes

Reviewer #2: Yes

3. Has the statistical analysis been performed appropriately and rigorously?

Reviewer #1: Yes

Reviewer #2: Yes

4. Have the authors made all data underlying the findings in their manuscript fully available (please refer to the Data Availability Statement at the start of the manuscript PDF file)?

Reviewer #1: Yes

Reviewer #2: Yes

5. Is the manuscript presented in an intelligible fashion and written in standard English?

Reviewer #1: Yes

Reviewer #2: Yes

6. Review Comments to the Author

Reviewer #1: Comments are adequately addressed

Reviewer #2: the comments are addressed
